# Coordination Study on Ecological and Economic Coupling of the Yellow River Basin

**DOI:** 10.3390/ijerph182010664

**Published:** 2021-10-12

**Authors:** Yanhong Zhao, Peng Hou, Jinbao Jiang, Jun Zhai, Yan Chen, Yongcai Wang, Junjun Bai, Bing Zhang, Haitao Xu

**Affiliations:** 1School of Earth Science and Mapping Engineering, China University of Mining and Technology (Beijing), Beijing 100083, China; zhaoyh@student.cumtb.edu.cn; 2Satellite Environment Application Center, Ministry of Ecology and Environment, Beijing 100094, China; zhaijunsec@163.com (J.Z.); chenyan30033@163.com (Y.C.); skdzyh@163.com (Y.W.); junj_bai@163.com (J.B.); zhangbing_1991as@163.com (B.Z.); xht0807@sina.com (H.X.); 3Chinese Research Academy of Environmental Sciences, Beijing 100012, China; 4College of Resource Environment and Tourism, Capital Normal University, Beijing 100048, China

**Keywords:** Yellow River Basin, economic development, ecological status, coupling degree, coupling coordination degree

## Abstract

The coupling and coordination relationship between ecology and the economy in the Yellow River Basin is a hot topic in sustainable development research. Said research has important guiding significance for the ecological security and comprehensive development of the Yellow River Basin. Taking the Yellow River Basin as the object of our study, based on the data of the economy, energy consumption data, ecology data and water resources data, we construct an indicator system of the economic development and ecological status of the Yellow River Basin and use the principal component analysis method to calculate the economic development and ecological status index. Then, we use the evaluation method, the coupling degree model and the coupling coordination degree model to analyze the time and space evolution trends of economic development and ecological state, coupling degree and coupling coordination degree. The results show that: (1) From 2000 to 2018, the economic development index of the Yellow River Basin rose steadily; the ecological status index showed a slow rise and then a downward trend. (2) The degree of coupling between economic development and ecological state has been considered as intensity coupling after 2005. The coupling trend slowly increased and then decreased, indicating that the interaction effect between the economy and ecology was first significantly enhanced and then slowly weakened. (3) The degree of coupling coordination increased from 0.2994 to 0.6266 and then decreased to 0.5917, reflecting the continuous improvement of the relationship between the regional economy and the ecological environment and the trend toward coordination. From 2015 to 2018, due to the gradual increase in the difference between economic development and ecological conditions, the coupling and coordination between the two decreased. Studies have shown that ecological construction and protection should be strengthened to ease the contradiction between the economy and ecology and achieve coordinated development.

## 1. Introduction

With the introduction of the concept of green development and the increase in national attention in this concept, how to realize the harmonious coexistence and common development of man and nature has gradually become a serious problem facing the world. Economic development directly reflects the degree of social development, but too fast economic development will inevitably cause ecological damage and resource consumption [1]. Therefore, the interaction between economic development and ecological conditions has become a research hotspot [2].

Scholars at home and abroad have conducted various studies on the interactions between economic development and ecology. Research has shown that only through economic and ecological principles, with trends of synergy, interaction, and simultaneous development [3], will sustainable development be achieved [4,5]. 

The existing research methods mainly analyze the relationship between economy and ecology from two aspects: quantitative and qualitative perspectives. Among them, the Environment Kuzz Curve, the Pressure-State-Response and constructing the corresponding indicator systems according to the actual problem are the main methods to evaluate the relationship of the economic and ecological environments. The Environment Kuzz Curve [6,7] analyzes the relationship between the economy and ecology from a qualitative point of view. The curve shows that ecology will first develop a trend of deterioration and then one of improvement with economic growth [8]. Yu and Lu [9] proved that the relationship between social economic factors and water environment pollution emissions was in line with the Environmental Kuznetz Curve model and quantified the two relationships using a linear model. However, the Environment Kuzz Curve only considers the trend of ecology with the economy but ignores the feedback effect of ecology on the economy [10]. The Pressure-State-Response (PSR) improved EKC’s feedback effects between economic and ecology to represent the interaction of them by pressure indicators, status indicators, and response indicators [11]. It is also commonly used to explain the interaction between economy and ecology [12]. The method of constructing the corresponding evaluation system according to the actual problem can evaluate the relationship between the economy and ecology from the perspective of quantitative and intuitive angles. It mainly includes the structural equation model [12], fuzzy analytic hierarchy process [5], coupling degree and coupling coordination degree models [13], etc. Among them, coupling degree and coupling coordination degree are two of the most commonly used methods, which can directly and quantitatively evaluate the degree of interaction and coordination relationship of two or more subsystems [14,15]. 

The analysis of inter-system coupling can be used in urban green sustainable development [16,17], carbon emissions [18,19], regional economic development [20], land and sea systems [21], disaster analysis [22], water environment [12], “Belt and Road” [2,9] and other areas. However, most of these studies only focused on a region or a city, and did not involve surrounding cities, and lacked a holistic analysis. In addition, the above study only used time-point data for analysis, without considering the steady-state situation of the coordinated changes in the study area in the time period.

The Yellow River is the second largest river in China and is the birthplace of the Chinese nation [23]. The evolution of the relationship changes in the ecological conditions of the Yellow River Basin and economic development had a huge impact on human production and life. Therefore, it is very necessary to explore the coordinated development relationship and evolution process between economic development and ecology in the Yellow River Basin. So, this paper selected the Yellow River Basin as the research object and prefecture-level cities as the data statistics unit to construct an indicator system for economic development and ecological conditions. The study also analyzed the economic development and ecological conditions of the Yellow River Basin from 2000 to 2018, as well as the degree of coupling between the two. The economic development system focused on the output value structure and resource consumption as the main evaluation parameters; the ecological status development system used the ecological structure and ecological quality as the main evaluation parameters.

The overall purpose of this study is to support the decision making in the sustainable development strategy and then provide guidance and suggestions for the ecological security and comprehensive development of the Yellow River Basin. It also provides technical support for managers and policy makers when studying the sustainable development and drawing up related policies for the Yellow River Basin.

## 2. Research Methods and Data Sources

### 2.1. Study Area

The Yellow River originates from the Bayan Har Mountains in Qinghai Province, China. It flows through 9 provinces including Qinghai, Sichuan, Gansu, Ningxia, Inner Mongolia, Shaanxi, Shanxi, Henan, and Shandong, as shown in Figure 1. In the analysis and research, the prefecture-level cities were selected as the data unit.

The research area does not completely cover the whole province. If the provincial data were used as the analysis unit, the deviation from the actual study area was too high to be able to represent the study area, which will bring errors to the results. Additionally, the availability of county-level data was poor. Moreover, the boundaries between prefecture-level cities were most consistent with the boundaries of the Yellow River Basin. Therefore, it is most reasonable to choose prefecture-level cities as the unit of data statistics and analysis, from the perspective of the availability of data and the coincidence of boundaries.

### 2.2. Data Sources

According to the source of the data, the data were divided into economic data, energy consumption data, ecological data and water resources data. The basic data used in the study included administrative boundary data and Yellow River Basin boundary data.

#### 2.2.1. Economic and Energy Consumption Data

Economic and energy consumption data mainly included energy consumption, Gross Domestic Product, and the proportion of each output value. The data were directly derived from the 2000 to 2018 statistical yearbooks of Qinghai, Sichuan, Gansu, Ningxia, Inner Mongolia, Shaanxi, Shanxi, Henan, and Shandong. The missing data were calculated based on the values in the statistical yearbooks of the 9 provinces mentioned above, such as the proportion of the first output value, the second output value, and the third output value that are missing in some provinces.

#### 2.2.2. Ecological Data

##### Ecological Composition Data

Land use classification data are necessary data for studying ecological composition. They were generated by artificial visual interpretation of Landsat TM/ETM remote sensing images and used to describe the type of land use data. The data format of the land use dataset was 30 m raster.

In order to make the land use data correspond to other data, the land use data of 2000, 2005, 2010 and 2015 represent the ecological compositions of 2000–2005, 2005–2010, 2010–2015 and 2015–2018 (Figure 2). Land use classification data had six primary types: cultivated land, forest land, grassland, water area, residential land and unused land. We reclassified the land use data primary class into agriculture, ecological and urban space, show in Figure 2. 

The areas and area proportions of agricultural space, ecological space and urban space were used to measure the composition of ecological space. We also selected the proportion of the natural ecological space area to measure the area of ecological space in the entire region in this study.

##### Vegetation Ecological Quality Data

Vegetation coverage was used to describe the coverage of vegetation in the area, which can effectively evaluate the ecological quality of the study area. It was calculated from the NDVI product data of MODIS (MODIS/Terra Vegetation Indices, MOD13Q1) based on the pixel binary model in this paper. The time resolution of these data is 16 days, the spatial resolution is 250 m, and the period is from 2000 to 2018.

#### 2.2.3. Water Resources Data

The total water consumption is the gross water consumption including the water transmission loss allocated to various users [24]. The total amount of water resources is the sum of surface water resources, groundwater resources, and groundwater evaporation, subtracting the recalculation amount of mutual conversion between surface water resources and groundwater resources [24]. The total water consumption and total water resources used in this article were derived from the Yellow River Basin Water Resources Bulletin and the water resources bulletins of the nine provinces from 2000 to 2018.

### 2.3. Method

#### 2.3.1. Economic and Ecological Status Coupling Assessment Ideas

Figure 3 shows the research framework system, which mainly includes data sources, index system, result analysis, etc. The research contents are divided into two parts: economic development and ecological conditions. This paper firstly constructed economic indicators and ecological status indicators, then calculated the economic development index and ecological status index, and finally quantified the coupling degree and coupling coordination degree of the two with the coupling degree and coupling coordination degree model.

#### 2.3.2. Index Selection and Weight Calculation

##### Indicator Selection

For the study area, the economic development index and ecological status index were selected for the coordinated study of the economic and ecological coupling of the Yellow River Basin. We evaluated the system of the Yellow River Basin into the economic development system and ecological status system. The systems were divided into first-level indicators and second-level indicators (Table 1).

##### Index Weight Calculation

Since the data may be correlated, principal component analysis was selected to reduce the dimensionality of the data and then calculate the index weight. The correlations between indicators were based on KMO and Bartlett spherical inspection statistics. According to the principle of the accumulated variance contribution rate being greater than 80%, the representative component of the economic development system and the ecological condition system was selected separately. From Table 2, the economic development system selected three main components; the ecological condition system also selected three main components. Then, the initial weight model coefficient was calculated, thereby calculating the weight. The characteristic value corresponding to the selected main component is greater than 1; the contribution rate is greater than 80%. The calculation process can be divided into the following steps: 

(1)Data standardization

Due to the difference in the measurement of each indicator data, the data needed to be standardized. In data standardization, the data were first divided into positive and negative indicators, and then the data were standardized using Formula (1).
(1)$$x_{i}^{\prime }=\left\{ \begin{array}{ll} \frac{x_{i}-\min(x_{i})}{\max(x_{i})-\min(x_{i})} & \;\mathrm{Positive indicators}\\ \frac{\max(x_{i})-x_{i}}{\max(x_{i})-\min(x_{i})} & \;\mathrm{Negative indicators} \end{array}\right.$$
where $x_{i}$ is the value of the second-level indicator; $x_{i}^{\prime }$ is the value after standardization.

(2)Weight calculation

Using the existing 12 original data $x_{i}$, the weights of the primary and secondary indicators in the study area were calculated by principal component analysis, as shown in Table 1.

#### 2.3.3. Evaluation of Economic Development and Ecological Conditions

In order to clearly judge the degree of development of the economic and ecological conditions, the multiple of the ratio of the economic and ecological conditions index was used to judge the level of development. The nodes for judging and evaluating were based on the scatter plot of the ratio of the ecological condition and the ecological condition index. It can be seen from Figure 4 that the scattered points of economic and ecological development were mainly concentrated in (0, 2). The threshold segmentation method of the mean method was adopted [1], and 0.5 times, 1 times and 1.5 times (Table 3) were determined as the nodes for determining the level of economic and ecological conditions in order to divide the degree of development into four levels. 

#### 2.3.4. Evaluation Method of Coupling Coordination Degree

Aiming to research the coupling correlation between economic development and ecological conditions in the Yellow River Basin, the coupling degree model and the coupling coordination degree model were used to analyze the correlation analysis between economic development and ecological conditions.

Coupling degree was used to analyze the phenomenon and degree of interaction between two or more systems. It can quantitatively analyze the degree of mutual influence of each system to a certain extent and predict the development trend of the system [5,13,17]. The greater the degree of coupling, the more orderly the development direction between the elements.

The degree of coupling coordination reflects the coordination quality and relationship of interactions between different systems, measures the degree of harmony between each system, and reflects the sustainability of the region [5,13,17].

The economic development/ecological status index, comprehensive development index and coupling degree were necessary parameters for calculating the coupling coordination degree. Therefore, the calculation of the coupling correlation between economic development and the ecological environment mainly started with the comprehensive development index, coupling degree, and coupling coordination degree. The specific calculation method is as follows:

(1)Economic Development Index:

(2)$$f(x)=\sum_{i=1}^{m}\lambda_{i}x_{i}^{\prime }$$f(x) is the Economic Development Index, $x=\lambda_{i}x_{i}\left(i=1,2,\cdots ,6\right)$; m represents the number of secondary indicators of the evaluation system; $\lambda_{i}$ is the economic development index’s weight.

(2)Ecological Status Index:

(3)$$g(x)=\sum_{i=1}^{n}\gamma_{i}x_{i}^{\prime }$$g(x) is the Ecological Status Index, $x=\lambda_{i}x_{i}\left(i=7,8,\cdots ,12\right)$; n represents the number of secondary indicators of the ecological status system; $\gamma_{i}$ is the ecological status index’s weight.

(3)Coupling degree calculation



(4)
$$C_{fg}=\sqrt{\frac{f(x)\cdot g(x)}{{\left(\frac{f(x)+g(x)}{2}\right)}^{2}}}$$



(4)Coupling coordination degree calculation

When calculating the coupling coordination degree, the comprehensive development index is required. The comprehensive development index is obtained from the economic development index and the ecological status index, and used to describe the changes and development levels of the system [5,13,17]. We calculated the comprehensive development index of economic and ecological as (5).
(5)$$T_{fg}=\alpha f(x)+\beta g(x)$$
α
and β represent the weight of the economic development system and the ecological state system, respectively. In order to make economic development and ecological development balanced, make α=β=0.5.

Coupling coordination degree was obtained by the coupling degree and the comprehensive development index, calculated as in Formula (6):(6)$$D_{fg}=\sqrt{C_{fg}\times T_{fg}}$$

The value of the coupling range was (0, 1); the closer to 1, the stronger the relationship between the economy and ecosystem. Conversely, the relationship between the economy and ecosystem is not obvious. The larger the value of the coupling coordination degree, the more coordinated the development of the economy and the ecosystem. The coupling degree and coupling coordination level [25] are shown in Table 4.

## 3. Result

### 3.1. Temporal and Spatial Changes in the Economic Development of the Yellow River Basin 

Figure 5 shows the economic development in the Yellow River Basin, as well as the development of the economic industrial structure and resource consumption subsystem. For a long time, the economic development index in the Yellow River Basin saw a rising trend, indicating that the economy was constantly evolving. Index of economic industry structure continued to increase with time; the resource consumption index increased rapidly in 2000 to 2010, slowly increased in 2010–2015, and slowly declined in 2015–2018.

Differences in the economic development of different provinces and the Yellow River Basin. Figure 6 was an economic development status index of various provinces in the study area. From 2000 to 2018, Gansu’s economic development index increased with time. Ningxia and Sichuan went through a trend of increasing and decreasing, increasing from 2000 to 2010, and then decreasing from 2010 to 2018. Inner Mongolia, Qinghai, and Shandong had an increasing trend in 2000 to 2015, and then a slowly reducing trend in 2015 to 2018. Shanxi and Henan had reducing trends in 2000 to 2010, which increased in 2010–2018; Shaanxi was had a reducing trend in 2000 to 2015, but in 2015–2018 this slowly increased.

The provincial economic development was divided into four levels, and the evaluation of the provinces is shown in Figure 6. Overall, the level of economic development in 2000 to 2018 continued to improve over time. Among them, the economic development of Qinghai and Sichuan had a low level of development. Shandong’s economic development level far exceeds other provinces and was in its highest level of development. Henan and Shanxi were also at a higher level of development.

With regard to the spatial distribution, the trend of economic development in the Yellow River Basin decreased from east to west. The economic development of the eastern part is significantly faster than the development of the west. The economy of the area where the Yellow River enters the sea saw high-level development. Figure 7 and Table 5 showed a significant difference in economic development with regard to space. In 2000–2018, economic development continued to increase, and the cities undergoing middle and mid to high economic development were mainly concentrated in the eastern part. The cities of economic development in 2000–2010 can reach 12, and there were 24 high-level cities. There were 18 cities that changed in 2010 to 2018; among them, 26 cities achieved high levels of development. This indicates that the economy in 2000 to 2018 was constantly evolving, while the development levels of the cities were constantly improving.

### 3.2. Temporal and Spatial Changes in the Yellow River Basin’s Ecological Status

From Figure 8, the ecological status index from the Yellow River Basin in 2000 to 2015 had an upward trend, and in 2015 to 2018 underwent the development trend of decline. For the ecological quality subsystem, 2000 to 2010 saw a decline in development, and 2010 to 2018 saw improvements. This shows that since 2010, the emphasis on the ecological quality of the study area had continued to improve the ecological quality significantly. 

For the ecological constitute subsystem, 2000 to 2015 saw the ecological index gradually increase, while in 2015 and 2018 the national spatial constitute index fell rapidly. That was because the ecological space area was occupied by urban space. Table 6 shows the ecological, urban and agricultural spatial area of 2000 to 2015. The ecological spatial area in 2015 decreased by 0.36% compared to 2010, and the urban spatial area increased by 4.6%. However, the proportion of ecological spatial area, compared to 2000, increased slightly in 2010, an increase of 0.09%, and the urban spatial area increased by 0.20%.

There were regional differences in the ecological status index from different provinces, as shown in Figure 9. Qinghai and Sichuan’s ecological statuses underwent the highest levels of development; the ecological status of Henan and Shandong were the worst, both experiencing low-level development. Gansu’s 2000–2005 ecological status was at the mid to high level, but in 2005–2018 it was at a medium level.

With regard to the spatial distribution, the ecological status of the Yellow River Basin was the opposite to economic development (Figure 10). The ecological status from east to west gradually improved. Among them, there were 20 high-level cities, 11 mid- to high-level cities, and 41 medium-level cities in 2000–2005 (Table 5). The number of high-level cities in 2000–2005, 2005–2010 and 2010–2015 were 20, 23 and 26, respectively. This shows that the ecological condition continued to recover. However, from 2015 to 2018, there were 22 high-level cities and 10 mid- to high-level cities, showing that the ecological status was more stable and developed. There was also a regional difference in the development of different prefecture-level cities.

### 3.3. Economic Development and Ecological Status Coupling Time and Space Change

In order to better explore the intrinsic relationship between the economy and ecology, we used formulas to calculate the integrated development index (Formulas (2) and (3)), coupling degree (Formula (4)), and coupling coordination degree (Formula (6)) of the economy and ecology to quantify the degree, interaction intensity and coordination quality.

Table 7 and Figure 11 show the trend of changes in the comprehensive development index, coupling degree and coupling coordination degree of 2000 to 2018. The integrated development index of the economy and ecology increased and then reduced. The two were mildly coupled in 2000 to 2005; in 2005 to 2018, the coupling was more than 0.9, which is considered intensity coupling. However, the trend of the coupling coordination degree was increased first and then it decreased. It was considered moderately maladjusted, barely coordinated, and primarily coordinated in 2000–2005, 2005–2010 and 2010–2015, respectively; then, in 2015 to 2018, it changed again to barely coordinated.

In 2000 to 2018, the economic and ecological coupling in Gansu, Henan, Inner Mongolia, Ningxia, Shanxi and Shaanxi was considered as intensity coupling; Shandong and Qinghai were considered moderately coupled. Sichuan’s coupling in 2010–2018 was moderately coupled, while in 2000–2010 it was considered as intensity coupling. Figure 12 can be seen. The coupling coordination level of Henan in 2000–2015 was on the verge of maladjustment, while in 2015–2018 it was barely coordinated. Inner Mongolia’s coupling level in 2000–2005 was on the verge of maladjustment; in 2005–2018, it was barely coordinated. Sichuan’s coupling coordination levels in 2000–2010 and 2010–2018 were barely coordinated and on the verge of maladjustment, respectively. 

In spatial distribution, the coupling of most prefecture-level cities in the Yellow River Basin was considered as intensity coupling (Figure 13, Table 5). Among them, there were 77 cities considered to be strongly coupled and one city that was mildly coupled from 2000 to 2010. In 2010–2015, the intensity coupled cities was 78. In 2015–2018, strongly coupled cities amounted to 76, whereas mildly coupled and moderately coupled cities equaled one. The difference in coupling of the cities is due to the development of economic and ecological were different, affecting the interaction of the economy and ecology.

Figure 14 and Table 5 show the coupling coordination degree spatial distribution of the economy and ecology in the Yellow River Basin. In space, the barely coordinated cities were mainly distributed in the north and south of the research area. From the number of cities, the barely coordinated cities in 2005 to 2010 increased by three from 2000 to 2005. This indicates that the degree of coupling and coordination between the economy and ecology in the study area has been continuously enhanced from 2000 to 2010. In 2010–2015, the barely coordinated cities were reduced to 10, but the cities on the verge of maladjustment were increased to 67. After 2015 to 2018, the barely coordinated cities were increased by four from the previous five years. There was spatial heterogeneity in the economic and ecological coupling coordination degrees of various cities in the study area.

## 4. Discussion

Due to its unique geographic location, the Yellow River Basin faced more severe environmental protection and economic development; therefore, promoting the coordinated development of its economy and ecology was a major issue related to sustainable development in the region. This article aimed to study whether the economic and ecological development of the Yellow River Basin was coordinated. We analyzed the development degree, coupling degree and coupling coordination degree of the economic and ecological development of the Yellow River Basin in the overall, provincial-level and prefecture-level city.

### 4.1. Differences at Various Scales

We analyzed the development of economic and ecological development, coupling degree, and coupling coordination degree of economic and ecological development. There was a significant difference in time and space.

At the time scale, there was a significant difference in coupling and coupling coordination levels in 2000 to 2018. The coupling degree in 2000–2005 was lightly coupled, and the coupling of 2005–2018 was intensity coupled; the coupling coordination of 2000–2005, 2005–2010, 2010–2015 and 2015–2018 is moderately maladjusted, barely coordinated, primarily coordinated, and barely coordinated (Table 8). The coupling and coupling coordination level in prefecture-level cities and the provinces also had differences in 2000–2018 (Figure 11, Figure 12 and Figure 14).

There was also a spatial heterogeneity in the coupling and coupling coordination levels on the spatial scale. The levels of coupling and coupling coordination had spatial heterogeneity in the full-stream scale, provincial scale and prefecture-level scale. For example, in 2000–2005, the coupling degree of the entire basin is moderately maladjusted; the coupling of provinces was mainly on the verge of maladjustment and barely coordinated; the prefecture-level cities were mainly moderately maladjusted, mild maladjustment, on the verge of maladjustment and barely coordinated (Table 8, Figure 11 and Figure 13). In summary, the scale of the analysis is inconsistent, and there is spatial heterogeneity in the results [26].

### 4.2. Economic and Ecological Development 

The change in coupling degree and coupling coordination degree in 2000 to 2018 showed the difference in coupling degree and coupling coordination degree of the economy and ecology. This result is similar to previous studies on the relationship between the economy and the ecological environment. On a long timescale, the coupling degree and coupling coordination of the economy and the ecological environment will gradually increase over time, but there may be a decrease in the later period [20].

The reason for this is that the ecological and economic development is inconsistent. There were two aspects that caused the difference in ecological and economic development.

First, the support of national policies is responsible for this difference, which made the economy in the study area continue to develop. There is a significant regional difference in economic development [27], and the difference in economic development in various regions is due to the support of government policies [26]. In other words, the government’s policy support is the main driving force for economic development. In this regard, the Chinese Government should increase the policy support of economic development, change the unreasonable financial system, and reasonably configure the proportion of the first, second, and third industries.

Second, the relationship between economic development and ecological development was inhibitory. A too fast economic development will inevitably destroy the ecology, and the ecology will restrict the economic development through a series of feedback forms such as natural disasters, environmental pollution, and resource shortages [28,29,30]. However, the ecology had a certain lag in response to the economic system [31]. In order to make the ecological environment improve and develop and accomplish the coordinated development of economic and ecological environments, we should practice the scientific discussion of China’s “lucid waters and lush mountains are invaluable assets” and strictly implement the Yellow River forbidden fishing policies. 

### 4.3. Experimental Design-Related Issues and Defects

In addition to the cause of the results of the analysis, we also discussed the time period selection and data integration issues in this paper.

First, we considered the time selection problem. In order to analyze the coupling coordination degree of the economy and ecology of the Yellow River Basin, we selected 2000 to 2005, 2005 to 2010, 2010 to 2015 and 2015 to 2018 as the time interval of the analysis. The selection of time interval was due to the five-year change trend over a period of time, which more intuitively reflected the fluctuations in time during the time period, so that the changes were more obvious. 

Second, we considered the integration of data. In addition to land use classification data, other parameters were used from 2000–2005, 2005–2010, 2010–2015 and 2015–2018, such as the mean of each time period to express the steady state changes of parameters in the time period. Due to the slowness of the land use classification data and the lag of statistics, we used the 2000, 2005, 2010 and 2015 land use data of the ecological compositions in 2000–2005, 2005–2010, 2010–2015 and 2015–2018 (Figure 2). 

All in all, data lags and the data integration process may bring errors to the results. These is also the limitations of our study.

## 5. Conclusions

This study combines the coupling degree and coupling coordination model and the coupling coordination of economic development and ecological status. Unlike most of the previous research, this paper introduces economic structural and energy consumption first-level indicators in the system of economic development, introducing ecological structural and ecological quality first-class indicators in the ecological status system. Based on these indicators, we analyzed the time and space evolution of economic development, ecological state and coupling and coupling coordinated degree in the Yellow River Basin from 2000 to 2018. The main research findings are as follows:(1)The economic composition index and the economic development index have showed an upward trend, but the energy consumption index showed a slight downward trend in 2015–2018. This shows that the dependence of future economic development on basic energy consumption is gradually reduced. The economic index rose at an accelerated rate, indicating that the level of economic development and constant improvements, which further promoted the coupling of the economy and ecology.(2)The ecological quality index has continued to rise since 2010, while the ecological state index and ecological composition index showed a downward trend in 2010–2018. It is indicated that ecological quality has continued to improve, but ecological composition still plays a leading role in evaluating the ecological state.(3)As far as the evaluation of the level of economic development and ecological conditions is concerned, the level of economic development is low in the west and high in the east; the level of ecological conditions is high in the west and low in the east. This shows that there are significant regional differences in economic development and ecological status grading.(4)During 2000 to 2018, the economic and ecological coupling state in the Yellow River Basin showed a strong interaction that subsequently slightly weakened. The system of economic development and ecological state have mutually promoted each other.(5)During 2000 to 2015, the coupling coordination degree of the economy and ecology of the Yellow River Basin continuously improved during the study period, but it was reduced in 2015 to 2018. The coupling coordination level experienced moderate maladjustment, mild maladjustment, endangered maladjustment, slight coordination and primary coordination. The decrease in the degree of coupling coordination is due to the uncoordinated development of economic and ecological conditions.(6)The coupling degree and coupling coordination degree of economic and ecological conditions differ between large-scale and small-scale regions, which shows that the coupling degree and coupling coordination degree of different spatial scales present spatial heterogeneity.

In 2000–2018, economic development continued to increase; the ecological condition increased and then reduced due to the effect of ecological composition. The development difference between the economic development system and ecological condition system caused time change differences in coupling and coupling coordination. Due to the difference in research scale, the economic development, ecological status, coupling and coupling coordinated degree is space heterogeneity.

The above results will help the policy makers in the Yellow River Basin formulate appropriate sustainable development measures and establish and maintain the balance between economic development and ecological status.

## Figures and Tables

**Figure 1 ijerph-18-10664-f001:**
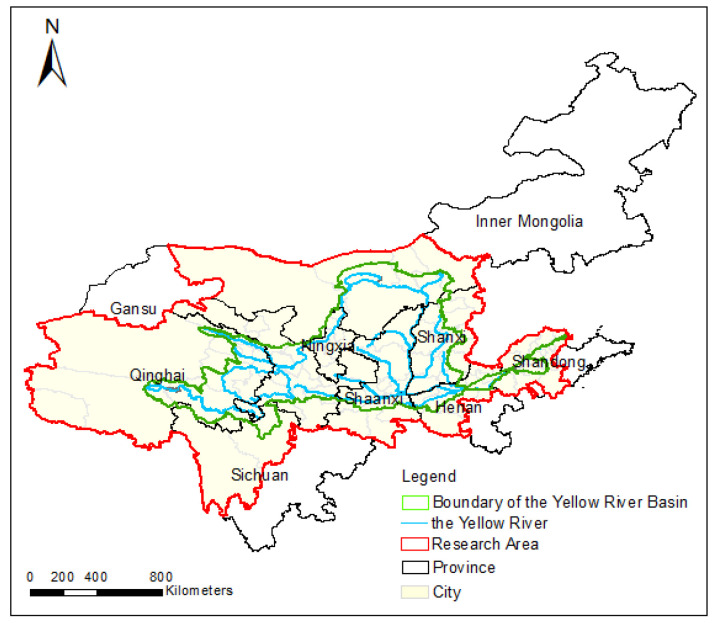
Overview of the study area.

**Figure 2 ijerph-18-10664-f002:**
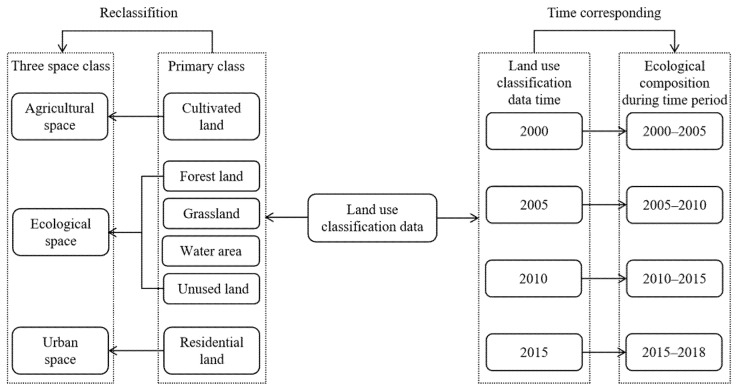
Land use data time correspondence and reclassification content.

**Figure 3 ijerph-18-10664-f003:**
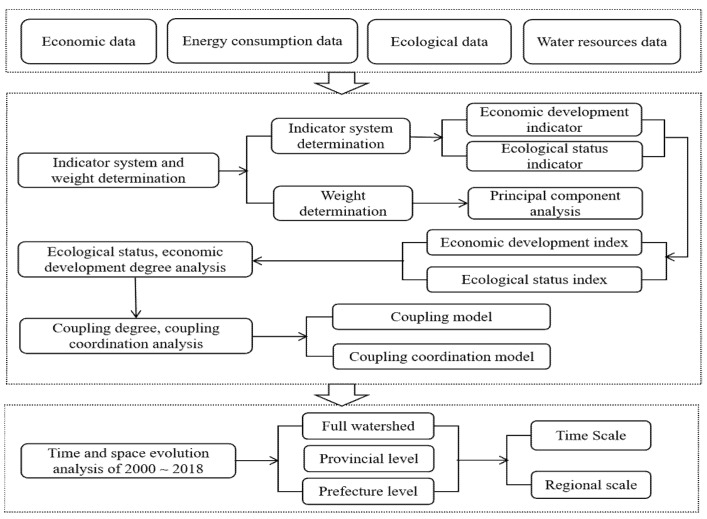
Research framework system.

**Figure 4 ijerph-18-10664-f004:**
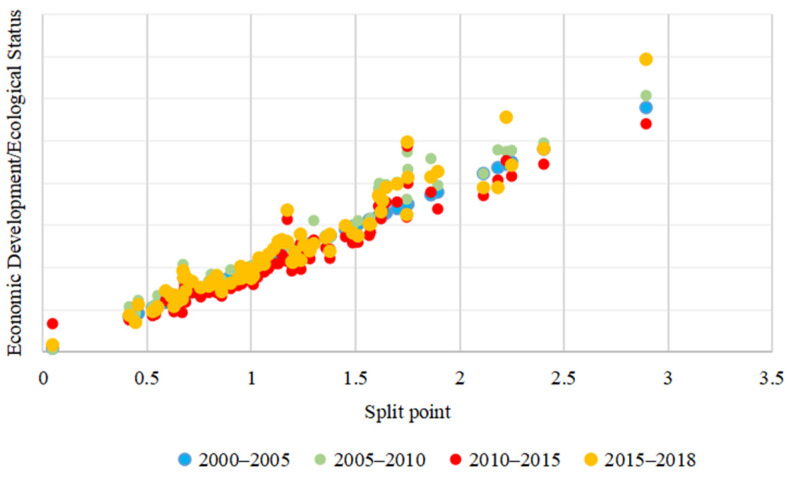
Scatter plot of economic and ecological development.

**Figure 5 ijerph-18-10664-f005:**
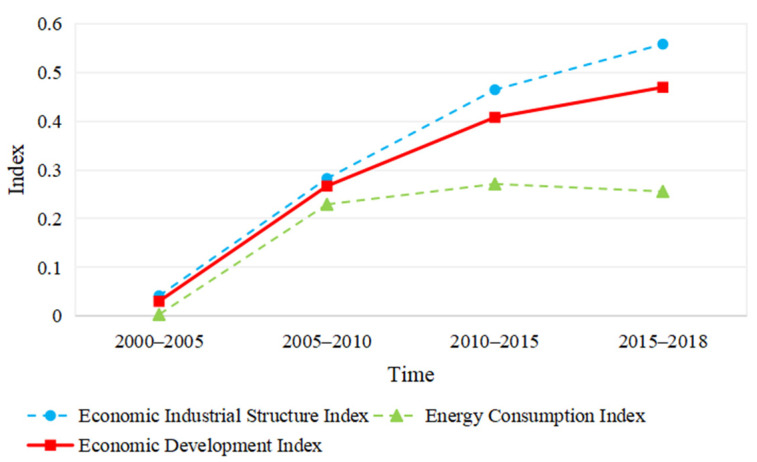
The trend of economic composite index of the Yellow River Basin.

**Figure 6 ijerph-18-10664-f006:**
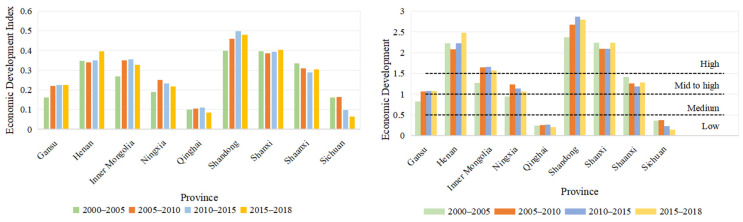
Economic development index and level of economic development of the provinces.

**Figure 7 ijerph-18-10664-f007:**
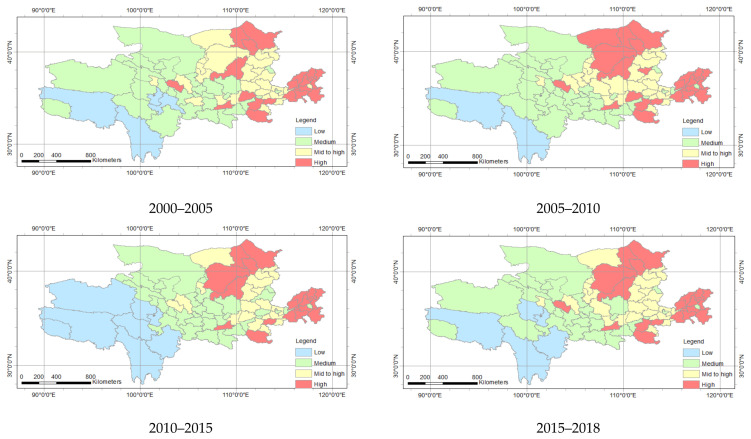
Spatial distribution of economic development in the Yellow River Basin.

**Figure 8 ijerph-18-10664-f008:**
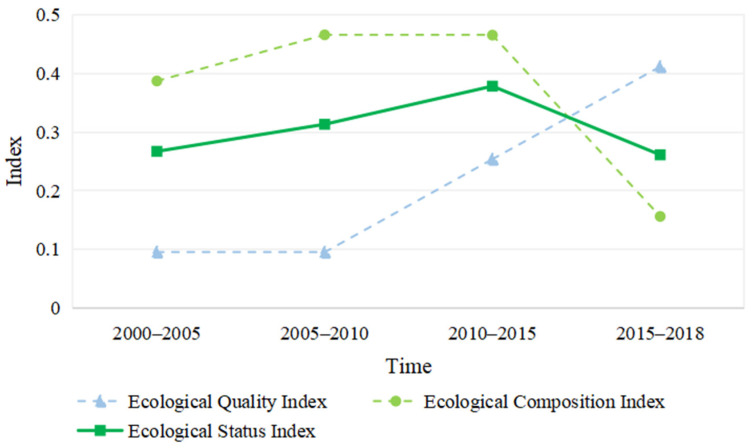
Change trend of ecological integrated index of the Yellow River Basin.

**Figure 9 ijerph-18-10664-f009:**
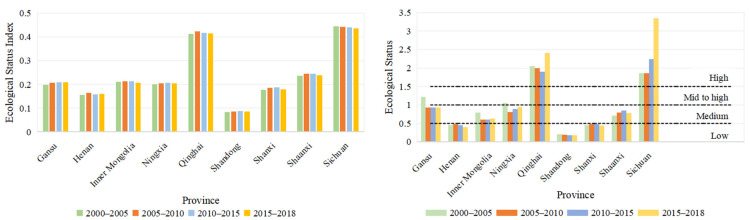
Ecological development status index and ecological status.

**Figure 10 ijerph-18-10664-f010:**
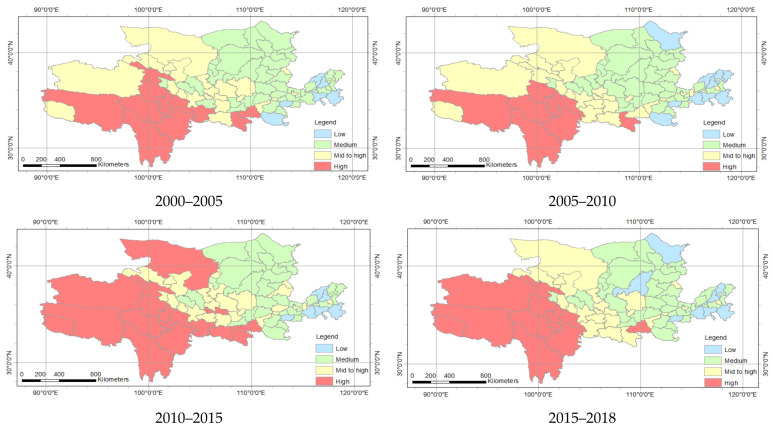
Space distribution of ecological status of Yellow River Basin.

**Figure 11 ijerph-18-10664-f011:**
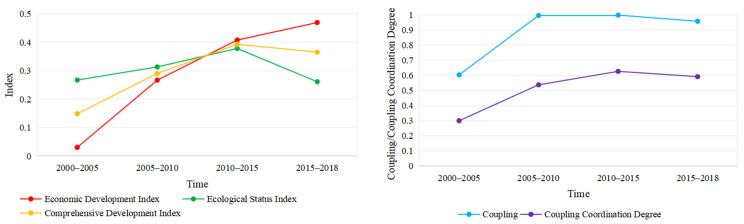
Economic and ecological status index and coupling relationship.

**Figure 12 ijerph-18-10664-f012:**
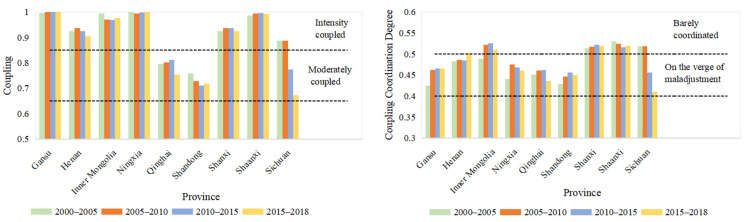
Changes of coupling and coupling coordination degree in provinces.

**Figure 13 ijerph-18-10664-f013:**
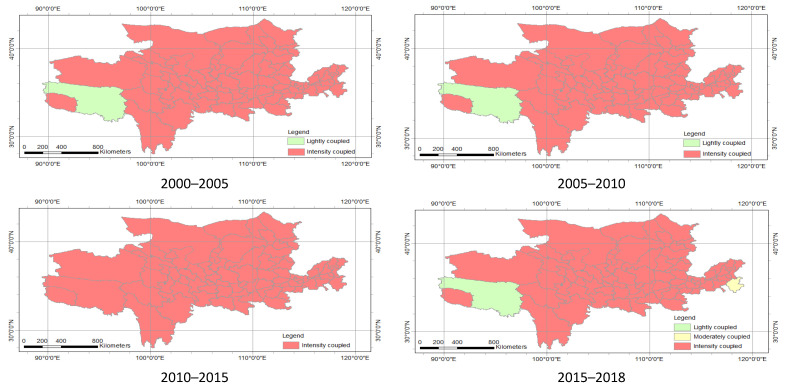
Space distribution of coupling degree of economy and ecology in the Yellow River Basin.

**Figure 14 ijerph-18-10664-f014:**
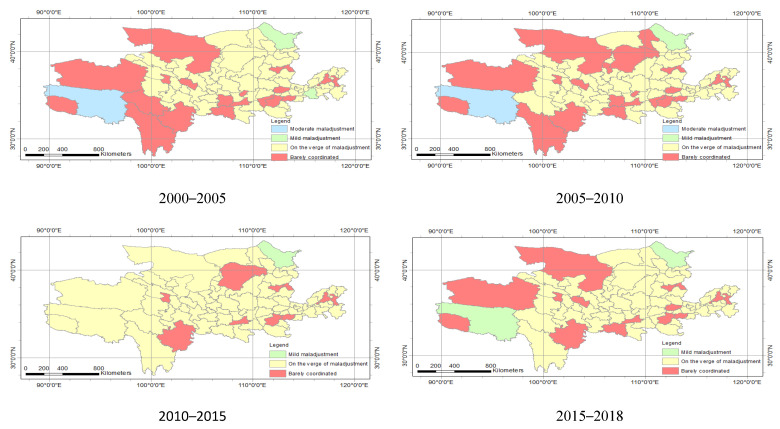
Spatial distribution of coupling coordination degree of the Yellow River Basin.

**Table 1 ijerph-18-10664-t001:** Comprehensive evaluation system and weights of the Yellow River Basin.

Evaluation System of the Yellow River Basin	First-Level Indicators	Weights	Second-Level Indicators	Indicator Attributes	Weights
Economic development system	Economic industrial structure	0.7070	Gross domestic product (*x*_1_)	+	0.2317
			Proportion of first output value (*x*_2_)	−	0.1184
			Percentage of second output value (*x*_3_)	+	0.1505
			Proportion of third output value (*x*_4_)	+	0.2064
	Resource consumption	0.2930	Energy consumption (*x*_5_)	+	0.1350
			Total water consumption (*x*_6_)	+	0.1579
Ecological status system	Ecological quality	0.4114	Vegetation coverage (*x*_7_)	+	0.2237
			Total water resources (*x*_8_)	+	0.1877
	Ecological composition	0.5886	Agricultural space area (*x*_9_)	−	0.1518
			Urban space area (*x*_10_)	−	0.1698
			Ecological space area (*x*_11_)	+	0.1774
			Percentage of ecological space area (*x*_12_)	+	0.0895

**Table 2 ijerph-18-10664-t002:** Eigenvalues and contribution rate of economic development system and ecological status system.

Economic Development System				Ecological Status System			
Element	Eigenvalues	Contribution Rate (%)	Cumulative Contribution Rate (%)	Element	Eigenvalues	Contribution Rate (%)	Cumulative Contribution Rate (%)
1	2.251	37.519	37.519	1	2.604	43.398	43.398
2	1.405	23.422	60.941	2	1.233	20.557	63.955
3	1.205	20.081	81.022	3	1.076	17.938	81.893
4	0.725	12.081	93.103	4	0.590	9.831	91.723
5	0.406	6.774	99.877	5	0.287	4.777	96.501
6	0.007	0.123	100	6	0.210	3.499	100

**Table 3 ijerph-18-10664-t003:** The level of economic development and ecological development.

The Ratio of Economic and Ecological	Economic Development	Ecological Status
(0, 0.5)	Low	Low
(0.5, 1)	Medium	Medium
(1, 1.5)	Mid to high	Mid to high
(1.5, +∞)	High	High

**Table 4 ijerph-18-10664-t004:** Coupling degree and coupling coordination degree level.

Coupling Interval	Coupling Level	Coupling Coordination Interval	Coupled Coordination Level
(0, 0.35)	Slightly coupled	(0.2, 0.3)	Moderate maladjustment
(0.35, 0.65)	Lightly coupled	(0.3, 0.4)	Mild maladjustment
(0.65, 0.85)	Moderately coupled	(0.4, 0,5)	On the verge of maladjustment
(0.85, 1)	Intensity coupled	(0.5, 0.6)	Barely coordinated
		(0.6, 0.7)	Primary coordinated

**Table 5 ijerph-18-10664-t005:** The level of economic development, ecological status, coupling degree and coupling coordination grading prefecture-level number statistics.

Level		2000–2005	2005–2010	2010–2015	2015–2018
Economic development	Low	4	2	7	5
	Medium	27	29	34	27
	Mid to high	27	23	18	20
	High	20	24	19	26
Ecological status	Low	6	10	6	8
	Medium	41	37	31	38
	Mid to high	11	8	15	10
	High	20	23	26	22
Coupling level	Sightly coupled	0	0	0	0
	Lightly coupled	1	1	0	1
	Moderately coupled	0	0	0	1
	Intensity coupled	77	77	78	76
Coupled coordination level	Moderate maladjustment	1	1	0	0
	Mild maladjustment	2	1	1	2
	On the verge of maladjustment	56	55	67	62
	Barely coordinated	19	21	10	14

**Table 6 ijerph-18-10664-t006:** The ecological, urban and agricultural spatial area proportion.

Area Proportion (%)	2000–2005	2005–2010	2010–2015	2015–2018
Agricultural spatial area proportion	17.12	16.90	16.84	16.51
Urban space area proportion	1.76	1.89	1.96	2.64
Ecological space area proportion	81.12	81.21	81.21	80.85

**Table 7 ijerph-18-10664-t007:** Coupling situation of economic and ecological development.

Time	Economic Development Index	Ecological Status Index	Integrated Development Index	Coupling	Coupling Coordination Degree
2000–2005	0.0301	0.2671	0.1486	0.6033	0.2994
2005–2010	0.2667	0.3132	0.2899	0.9968	0.5376
2010–2015	0.4078	0.3781	0.3930	0.9993	0.6266
2015–2018	0.4694	0.2612	0.3653	0.9585	0.5917

**Table 8 ijerph-18-10664-t008:** The coupling degree and coupling coordination level of the Yellow River Basin.

The Level of Coupling and Coupled Coordination	2000–2005	2005–2010	2010–2015	2015–2018
Coupling level	Lightly coupled	Intensity coupled	Intensity coupled	Intensity coupled
Coupled coordination level	Moderate maladjustment	Barely coordinated	Primary coordination	Barely coordinated

## Data Availability

The data set is provided by Data Center for Resources and Environmental Sciences, Chinese Academy of Sciences (http://www.resdc.cn/), date accessed in 2 October 2021.

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
