# Peer review of "Coordination Study on Ecological and Economic Coupling of the Yellow River Basin"

_ijerph, 2021, doi:10.3390/ijerph182010664_

Round 1

Reviewer 1 Report

It is an interesting investigation (economic and social development VS ecological or environmental development). The entire process of integration of information and application of mathematical instruments are correct. The results are correct. The only detail is the high analysis of these results. Not only should the findings, the conclusions show thelogors achieved by the investigation, are not descriptions (line 414-417), the results must be analyzed to generate the confusements and nine should not be descriptions (line 418-429). It is recommended that the work developed be convened properly. "What is this study for this study?", "In what moment, this model of regional information analysis can be used?", which institutions can use it? 

Reviewer 2 Report

Dear Authors,

The subject of the research is important and interesting but in this form, paper has a few issues which should be clarified before acceptance.

Introduction part should be improved. Some aspects of the literature review must be strengthened. The research goal should be at the end of the introduction. The marking in Figure 6 should be in English. The discussion should also show the results of other authors. There is no discussion on results in comparison to relevant studies.

Reviewer 3 Report

The work is highly interesting but the method is unclear. The explanation of the method used must be reinforced to give weight to the results and analysis.  Some examples to clarify:

  • The author refers to a PCA, which I have neither seen nor perceived the utility of. I understood that it was to evaluate the weight of the variables. Are the calculated principal components used for the continuation? Were the most representative components selected to then select the weight of the variables in the construction of the dimensions? but if so, what single variance was set? There is a lot of information missing about this possible PCA.
  • In equations (1), (2) and (3), what does x represent? You can't just define f(x), you have to define x.
  • line 123 to 131: an explanation by a diagram would be more pleasant for the understanding.
  • Line 173 : “then the data was standardized using data standardization 173 formulas. “ : What is the standardization method used?
  • Figure 3 and related explanations should be expanded.
  • Line 304 : “There was a significant difference “ : There is a difference, but to determine its significance, we would need a significance index. Here the vocabulary is inappropriate.
  • Line 320 : “we used formulas to calculate “ which one?

Other remarks:

  • Legend Figure 11 : do you mean intensitly coupled?
  • Legend Figure 13 : Be careful with capitalization

I think that the conclusion must be strengthened as well. For example, nothing is said about the spatial differences found.

Round 2

Reviewer 3 Report

The author has responded to a majority of my comments and corrected them in order to improve the understanding of the work. Thank you for all these modifications. 

Some remarks remain nevertheless:

- On the conclusion: 1) the first paragraph is very unclear, and I think it needs to be reworked. Maybe there are even some grammatical errors? Be careful to write "paper" and not "papre"; 2) I think that a last paragraph of general conclusion is missing. 
- On PCA: the KMO and Bartlett indexes allow to determine if it is interesting to factor the data. In short, it is almost a preliminary step to PCA. So I'm still lacking information on PCA. It is used to calculate the weight, but which element of the PCA result was used? At this level of information given by the article, I understand the following: only one factorial axis is retained and the weights correspond to the weights of the variables in the calculation of dimension 1. This may not be the case at all, but I am still missing elements.

Author Response

This manuscript is a resubmission of an earlier submission. The following is a list of the peer review reports and author responses from that submission.